# Early Diagnostics of Vulvar Intraepithelial Neoplasia

**DOI:** 10.3390/cancers14071822

**Published:** 2022-04-04

**Authors:** Vesna Kesić, Pedro Vieira-Baptista, Colleen K. Stockdale

**Affiliations:** 1Medical Faculty, University of Belgrade, 11000 Belgrade, Serbia; 2Clinic for Obstetrics and Gynecology, University Clinical Center of Serbia, 11000 Belgrade, Serbia; 3Lower Genital Tract Unit, Centro Hospitalar de São João, 4200-319 Porto, Portugal; pedrovieirabaptista@gmail.com; 4Hospital Lusíadas Porto, 4200-319 Porto, Portugal; 5Department of Obstetrics & Gynecology, Vulvar Vaginal Disease and Colposcopy Clinics, University Iowa Healthcare, Iowa City, IA 52242, USA; colleen-stockdale@uiowa.edu

**Keywords:** vulva, precancer, diagnostics, vulvoscopy

## Abstract

**Simple Summary:**

The spectrum of vulvar disorders is wide and varies from infections, dermatoses, manifestations of hormonal and systemic conditions to vulvar intraepithelial neoplasia (VIN) and invasive cancer. It is not always possible to distinguish vulvar lesions on the basis of macroscopical aspects and the distribution of changes. For definite diagnosis of a vulvar lesion, a biopsy is needed. However, in practice, the decision to perform a biopsy is often delayed due to a lack of specificity of symptoms at the early stages of the neoplastic disease. The aim of this article is to provide clinicians, both gynecologists and dermatologists, with the main features of vulvar precancerous lesions, in order to recognize and treat them on time, thus preventing vulvar cancer. Clinical appearance of VIN is variable with significant variations present in color, surface, and topography. Evaluation of all VIN lesions should be conducted very carefully, because an underlying early invasive squamous cancer appears to be present in a significant percentage of patients.

**Abstract:**

The spectrum of vulvar lesions ranges from infective and benign dermatologic conditions to vulvar precancer and invasive cancer. Distinction based on the characteristics of vulvar lesions is often not indicative of histology. Vulvoscopy is a useful tool in the examination of vulvar pathology. It is more complex than just colposcopic examination and presumes naked eye examination accompanied by magnification, when needed. Magnification can be achieved using a magnifying glass or a colposcope and may aid the evaluation when a premalignant or malignant lesion is suspected. It is a useful tool to establish the best location for biopsies, to plan excision, and to evaluate the entire lower genital system. Combining features of vulvar lesions can help prediction of its histological nature. Clinically, there are two distinct premalignant types of vulvar intraepithelial neoplasia: HPV-related VIN, more common in young women, multifocal and multicentric; VIN associated with vulvar dermatoses, more common in older women and usually unicentric. For definite diagnosis, a biopsy is required. In practice, the decision to perform a biopsy is often delayed due to a lack of symptoms at the early stages of the neoplastic disease. Clinical evaluation of all VIN lesions should be conducted very carefully, because an underlying early invasive squamous cancer may be present.

## 1. Introduction

The spectrum of vulvar lesions is a complicated and challenging group of disorders, beyond that of extragenital skin, which vary from infections, dermatoses, effects of hormonal, and systemic disturbances to vulvar intraepithelial neoplasia and invasive cancer.

It is not always possible to distinguish between various types of vulvar lesions on the basis of macroscopical aspects and the distribution of vulvar changes. Distinction based on the characteristics of vulvar lesions is often not indicative of histology. While biopsies are not needed in most cases, the general rule in adult women, when there is therapeutic failure or a suspected premalignant or malignant condition, is to perform biopsy for definite diagnosis of a vulvar lesion. However, in practice, the decision to perform a biopsy is often delayed due to a lack of specificity of symptoms at the early stages of the neoplastic disease.

## 2. Terminology of Vulvar Intraepithelial Neoplasia

The concept of vulvar intraepithelial neoplasia has been accepted for more than a decade. In the past, an inappropriate parallel was drawn between cervical intraepithelial neoplasia (CIN) and VIN. Recent knowledge about the natural history of VIN has broadened, providing more information about the epidemiology, pathology, and clinical management of these diseases. There is no confirmation that the morphological spectrum of VIN 1–3 reflects the biological continuum, nor that VIN 1 is a cancer precursor [1]. This is why the term VIN 1 is referred to as a low-grade squamous intraepithelial lesion. On the other hand, good clinical–histological agreement has been achieved when VIN 2 and VIN 3 are combined in one diagnostic category. These two entities are today described as a high-grade vulvar squamous intraepithelial lesion. The squamous terminology of the lower anogenital system introduced in 2012 (lower anogenital squamous terminology—LAST) [2] and the WHO histological classification of Tumors of Female Reproductive Organs published in 2014 [3] classify the atypia of the squamous epithelium in different localizations in a uniform way. On the basis of these divisions, the ISSVD updated its VIN classification in 2015, considering the LAST/WHO terminology [4]. Today, vulvar squamous intraepithelial lesions are categorized as follows:Vulvar low-grade squamous intraepithelial lesion—LSIL, which includes VIN1, flat condyloma, and human papilloma virus effect (subclinical HPV infection);Vulvar high-grade squamous intraepithelial lesion—HSIL (encompassing VIN2 and VIN3), previously referred to as “intraepithelial vulvar neoplasia of the usual type”;Differentiated type vulvar intraepithelial neoplasia—dVIN (not included in the LAST guidelines).

This terminology is used in this study while also citing the old classification when quoting papers that did not use the new terminology.

## 3. Anatomy and Histology of Vulva

From a diagnostic point of view, the vulva includes not only the external female genital organs, but also the external urethral orifice, the perineum, and the perianal area [5]. These structures differ embryologically and, consequently, histologically. While the vulval epithelium and its appendages are of ectodermal origin, the dermis originates from the mesoderm. The only structure derived from the endoderm, such as the bladder and urethra, is the vestibule.

The appearance of the lesion largely depends on the tissue structure requiring knowledge of the histology of vulval skin at the affected site. The pubic mound, lateral parts of the major lips, and the perianal area are covered by hair-bearing skin. The inner parts of the major and the entire minor lips and clitoris are covered by non-hair-bearing skin. The vestibule is covered by a non-keratinized squamous epithelium and contains mucus-secreting glands.

The vulva is responsive to sex steroids and hormonal activity during puberty, sexual intercourse, pregnancy, delivery, menopause, and the postmenopausal period [6]. Recognition of these cyclical hormonal effects is of high importance for the diagnosis and treatment of vulvar disorders.

## 4. Diagnostics of Vulvar Intraepithelial Neoplasia

Features of vulvar intraepithelial lesions differ. Pruritus is usually the dominant clinical symptom, although the vulvar intraepithelial lesions are symptomless in many cases. On the other hand, pruritus is not specific and is an accompanying symptom in many other vulvar disorders [7]

Cytology is less reliable for vulvar lesions than for cervical and vaginal atypia. The presence of a thick keratin layer that covers the vulvar epithelium and suboptimal cellularity in vulvar smears can explain cytological misdiagnosis. The use of a spatula end, for cytology collection in women with biopsy-proven (pre)malignancies, results in only 32% smears significant for VIN or vulvar carcinoma [8]. Various techniques for vulvar cytology have been described such as vulvar brush cytology [9], but even those did not yield adequate results because of scarce cellularity. Vulvar cytology is currently not recommended.

A more reliable diagnostic tool for examination of the vulva is vulvoscopy. Vulvoscopy is not the colposcopic examination of the vulva. This term is retained in clinical practice, but it should be intended as a composite diagnostic act involving careful naked-eye and low-power magnified examination when needed. It can be performed by not only a gynecologist, but also any physician specifically skilled in vulvar disease [10].

In most cases, significant vulvar lesions are easily recognizable by naked eye. Magnification can be achieved using a magnifying glass or a colposcope. Due to the normal histology of this area, which is covered by a keratinized, stratified squamous epithelium, the use of a colposcope is not as beneficial in defining the nature of vulvar lesions as when used for the cervix [11]. However, it can be useful for identifying the individual components of the lesions, for both biopsy and treatment purposes. Studies have confirmed the elevated prevalence of anal intraepithelial neoplasia in women with HPV-related precancer and cancer [12]. If an evaluation of the anal canal is needed, the patient should be referred to a clinician with adequate training for this procedure [13].

### 4.1. Tissue Basis of Vulvoscopy

The features and the structure of the vulvar tissue, such as the thickness of the epithelium, the vascularity of the underlying stroma, and the presence or absence of hair, determine the presentation of different vulvar lesions. Thickness of the skin varies between different areas of the vulva, which is why histologically identical lesions may have different appearance. Pigmentation can obscure blood vessels. Therefore, vascular patterns are less marked and less reliable than in colposcopy of the cervix. Vascular patterns such as punctations and mosaic changes do not easily develop on vulvar skin [10]. They are less common and practically can only be seen in the non-hair-bearing areas. These are the inner portions of labia minora where the keratin layer is thinner and the vestibular epithelium which does not contain a keratin layer. Thus, white plaques and an aceto-white epithelium are the most frequent colposcopic manifestations of vulvar pathology.

Stromal changes that influence colposcopic appearance are usually due to the increase in vascularity. This increase may be due to inflammation, an immune response, or the neovascularization of neoplasia. In these cases, the color of the skin will become red. Vascularization may also be decreased, or the stroma may undergo fibrotic changes, resulting in whitish coloring of the skin.

### 4.2. Technique of Vulvoscopy

Vulvoscopy is not a colposcopic examination of the vulva, but the technique does not differ from the usual colposcopic examination of the cervix. The patient is examined in the lithotomy position. All parts of the vulva have to be examined: labia majora and minora, vestibule, clitoris, terminal urethra, perineum, and perianal area. The multifocal nature of vulvar intraepithelial disease makes the examination more difficult.

In cases where high-grade lesions or lesions extending to the anus are detected, further exploration of the anal canal is recommended (Figure 1). The prevalence of anal HSILs among all women with proven vulvar HSILs is 11.8% to 18.2% [12,14]. Screening modalities for anal HSIL include anal cytology, standard anoscopy, or high-resolution anoscopy (HRA), which is performed with the aid of colposcope [15].

Examination of the vulva has several stages.

#### 4.2.1. Visual Examination

The examination of the vulva should start by visual examination (without magnification) of the entire vulvar region. The attempt should be made to clearly visualize the hair-bearing skin. Proper examination requires separation of the labia majora and minora and exposure of the entire vestibule. During this phase of the examination, areas of redness, thickness, pigmentations, ulcers, and atrophy must be sought. Genital warts or invasive cancer can be easily recognized.

If magnification is needed, the lowest magnification (6×) is used to quickly scan the vulva. Later, higher magnifications can be used, if necessary, to examine for smaller satellite lesions. Keratosis aggravates the normal opacity of the vulvar surface. In such cases, the magnification afforded under good lightning aids in the delineation of lesions. Application of physiological saline solution may decrease the keratinizing effect to a certain extent and assists in the visualization of abnormal vessels.

#### 4.2.2. Application of Acetic Acid

An integral part of all examinations using a colposcope is the application of acetic acid. Compared to the examination of the cervix, acetic acid has a less prominent effect when applied to the vulva, due to keratinized skin. This is why acetic acid should be more concentrated (5%) than when used for the cervix (3%) and must be applied more often and more abundantly. The application has to be long enough, usually 2–3 min, to allow vulvar lesions to show. It should be, however, remembered that this reaction is not specific, and that liberal use of acetic acid may be misleading in provoking aceto-white reactions other than vulvar neoplasia [10]. Aceto-whitening of the vulva has high sensitivity (97%) but low specificity (40%) as a predictor of high-grade vulvar intraepithelial neoplasia [16]. However, when the lesions are present, the application of acetic acid is useful to identify their extent and exact position. The absence of aceto-white lesions can reassure the clinician that high-grade vulvar lesion is absent, with a negative predictive value of 98% [16].

#### 4.2.3. Collins Test

The use of toluidine solution in the vulva is known as the Collins test, but it is no longer recommended due to its very low specificity [17].

#### 4.2.4. Biopsy

Vulvoscopy usually cannot predict the histological nature of the lesion in some cases, but it can help localize the best location(s) for biopsy(ies). Most vulvar lesions do not have specific characteristics. Proliferated tissue with an abnormal vascular pattern is always suspicious of invasion.

Biopsy is mandatory in the following cases:Fast growing lesions;Ulcers with >1 month of duration;Fields of bleeding;Each suspicious field of any color;Areas of induration;Lesions resistant to local medical treatment;Failure of previous therapy.

Biopsy should be planned very carefully and targeted to the areas of most prominent changes. Underdiagnosis is not unusual in preoperative biopsies of vulvar HSILs (VIN 2/3), accounting for 44.2% in cases of VIN 2 and 11.9% in cases of VIN 3 lesions. The reported rate of occult invasive cancer after local excision is 3.8% and 11.9%, respectively [18] (Figure 2). At initial treatment of vulvar HSIL (VIN3), an underlying squamous vulvar cancer may be found in 22% of patients [19].

## 5. Abnormal Vulvar Findings

Until recently, there has been no attempt to systematize colposcopic findings on the vulva. When attempting to classify vulvar findings, it is possible to use the same descriptive process for the vulva as for the cervix. Currently, the classification proposed by the International Society for the Study of Vulvovaginal Disease and the International Federation for Colposcopy and Cervical Pathology is in use [20]. (Table 1).

Combining particular features of vulvar lesions can help; however, in general, the prediction of the histological nature of vulvar lesions is less reliable than colposcopic grading of cervical lesions. The most informative characteristic of the lesion is probably color. It is easily visible to naked eye and may vary from white to black, depending on the pigmentation, vascularity of the dermis, and thickness of the overlying epithelium [21].

Vulvar lesions can be divided simply on the basis of skin color into skin-colored, red, white, or dark.

### 5.1. Skin-Colored Lesions

In normal conditions, the light reflects from the superficial blood vessels of the dermis, resulting in a pink color. Because vulvar intraepithelial lesions alter the architecture of the epithelium, they are not skin-colored.

### 5.2. Red Lesions

Any decrease in the thickness of the epithelium or any increase in the vascularity will give a red appearance. Thinning or ulceration of the epidermis, as well as inflammatory vasodilatation, an immune response, or neovascularization of neoplasia, will result in the appearance of red color. By visual examination, vulvar intraepithelial neoplasia may also present as red lesions, requiring the application of acetic acid to be used for better evaluation of the finding (Figure 3a,b).

Red lesions may present a local immune response or inflammatory reaction in conditions such as the following (Figure 4, Figure 5, Figure 6 and Figure 7):Inflammation (dermatitis, eczema);Infection (candidiasis, intetrigo, folliculitis);Dermatosis (psoriasis, lichen planus);Neoplasia (VIN, Paget’s disease, cancer).

### 5.3. White Lesions

White lesions are not always neoplastic. A superficial keratin layer, any degree of depigmentation, relative tissue avascularity, and/or reaction to acetic acid can contribute to the development of white color.

White or gray color can be the consequence of increased moisture of the vulval area, which causes maceration of keratin. The effect is more expressed if the keratin layer is thicker.

The loss or absence of melanin will lead to depigmentation. This can be seen in vitiligo when melanocytes in the basal layer are lost or destroyed, or when these cells have lost their ability to produce melanin.

Transient loss of pigment in a residual scar after healing of a skin ulcer may result in localized white lesions.

Decreased vascularity due to narrowing of superficial blood vessels as seen in lichen sclerosus may also give a whitish appearance.

White lesions may histologically be non-neoplastic epithelial disorders, HPV infection, or VIN (Figure 8, Figure 9 and Figure 10).

### 5.4. Dark Lesions

Dark lesions are due to an increased amount or concentration of melanin or blood pigment. Vulvar lesions appear dark if melanin is present intra-epithelially and/or intradermally. In these lesions, synthesis of melanin is amplified in epidermal melanocytes. The excess of melanin is subsequently ejected in the papillary dermis, and then taken by melanophages via the process of phagocytosis. This mechanism is known as “melanin incontinence” and produces the pigmented appearance of many vulvar intraepithelial lesions [22].

Dark lesions on the vulva as the result of increased pigmentation may occur after trauma, application of estrogen cream for the treatment of vaginal atrophy, or use of hormonal contraception.

The conditions where dark lesions may be seen are pigmentation disorders (hyperpigmentation), nevi, lentigo, and seborrheic keratosis, VIN, or malignant melanoma (Figure 11, Figure 12, Figure 13 and Figure 14).

Vulvar lesions may show a variety of color. Particularly recognizable are lesions which originate from vascular tissue such as angiomas or choriocarcinomas, which are typically violet. Necrotic tissue usually has yellow coloring.

IFCPC terminology of the vulva also defines lesions on the basis of secondary morphology presentation (Table 2).

Dermatological conditions such as allergic reactions present on the keratinized skin such as eczematous lesions (Figure 15), subclinical HPV infection, and vulvar intraepithelial lesions localized in the thickness of the epithelium only, particularly on the skin of the labia minora or vestibular epithelium, can be presented by a change in color (Figure 16).

Lichenification is the thickening of the tissue and loss of the usual reticulate, and it is often the result of constant scratching or rubbing. It is not a primary condition or disease, but rather a result of some underlying cause. Conditions that lead to lichenification (Figure 17) are numerous, ranging from dermatitis, psoriasis (Figure 18), and dry skin to trauma to the skin, stress, or anxiety disorders. Normal skin markings, such as cracks, wrinkles, or scales are exaggerated, giving a leathery or bark-like appearance to the skin.

Erosions and ulcers are below the level of the surrounding epithelium. They are typical findings for some infections and dermatoses such as herpes infection, syphilis, Behçet’s or Crohn’s disease, and other ulcerative and bullous skin disorders (Figure 19 and Figure 20). Ulcerative lesions may also suggest a granulomatous sexually transmitted disease or cancer.

It is not possible to distinguish between various types of lesions to predict their histological nature only on the basis of macroscopical appearance and the distribution of vulvar changes [23]. The coexistence of morphologically diverse vulvar skin lesions may cause difficulties with diagnosis and the selection of adequate treatment [24]. Therefore, biopsy is essential for definite diagnosis of a vulvar lesion. Vulvoscopy is very important for exact localization of the lesion, directing the biopsy site, and mapping its borders when planning excision.

## 6. Specific Features of Vulvar Intraepithelial Neoplasia

Vulvar intraepithelial neoplasia can appear anywhere in the vulva, including the labia majora and minora, posterior fourchette, perineum, and periclitoral and perianal area. The perianal area and anal canal may be involved particularly in cases in which the posterior part of the vulva is involved.

There is no single pathognomonic sign of vulvar intraepithelial neoplasia. Clinical features are variable with significant differences in number, size, shape, color, surface, thickness, and topography. Lesions may be solitary or more often multiple. They are characteristically papular, raised above the level of surrounding skin with sharp borders and a keratotic, roughened surface. Their color may range from white to red, gray, blue, or brown. An underlying early invasive squamous cancer, according to different reports, appears to be present in 11% to 22% of patients [19,25]. In a study of 186 patients who underwent local excision for vulvar HSIL (VIN2 and VIN3), the occult cancer rate was 3.8% and 11.9%, respectively [18]. This is why accurate biopsy mapping is of the utmost importance.

Clinically, there are two distinct premalignant forms of vulvar intraepithelial neoplasia: high-grade squamous intraepithelial lesion (vulvar HSIL) and differentiated type VIN (dVIN).

High-grade squamous intraepithelial lesion (HSIL) (formerly termed VIN usual type—uVIN, Bowenoid papulosis) tends to occur in young women and is associated with HPV infection. Genital HPV infection in women is predominantly acquired in adolescence, with peak prevalence in women aged 25 years or less, and a decrease in older age groups, particularly after the age of 35 [26]. Risk factors include smoking, number of sexual partners, and immunosuppression, as well as previous treatment of vulvar HSIL and a history of HSIL of the cervix or vagina. The rate of progression to invasive cancer has been reported to range from less than 5% [27] to 9.7% in women younger than 35 years treated for vulvar HSIL [28].

The lesions are typically multiple, discoid, white, or pigmented, and they can affect the whole external genital organs, including the perianal skin (Figure 21 and Figure 22). Some pigmented forms of vulvar HSIL have a specific clinical appearance described as flat gray–brown lesions, hyperpigmented raised lesions, and papillary or verrucous lesions.

Vulvar HSIL is most often multifocal and multicentric. Multicentric lesions (of the cervix, vagina, vulva perianus, or anal canal) are often present in cases with HSILs, underlining that HPV infection may involve squamous epithelia from the cervix to perianal area. The concurrent presence of high-grade squamous intraepithelial lesions at two or more of these sites is defined as “multizonal anogenital disease” [29]. Smokers and women on immunosuppressive drugs are at increased risk of multizonal anogenital disease (RR 1.84 and 2.57, respectively).

VIN differentiated type (dVIN) is not associated with HPV infection or smoking. It is less common than vulvar HSIL (up to 5% of all vulvar intraepithelial neoplasia), seen primarily in older women, usually in a field of lichen sclerosus (LS) or lichen planus. This type of VIN poses a high risk for rapid progression to vulvar squamous cell carcinoma. If untreated, dVIN will progress to squamous cell carcinoma (SCC) more often than HSIL (50% versus 5.7%) [30,31]. The absolute risk for primary VSCC in dVIN varies between 33% and 86%, with the median time for progression of 9–23 months [32].

It was shown that the LS-mediated pathway leads to dVIN, resulting in keratinizing SCC [33,34]. Close surveillance of women with lichen sclerosus is needed. It is conceivable that various, heterogeneous environmental factors acting on a genetic background trigger an autoimmune Th-1 response, which leads to a chronic inflammatory state [33]. LS affects women of all ages and often goes unrecognized and underreported [35]. Studies have also described lesions of uncertain malignant potential adjacent to dVIN and SCC, qualifying them as “vulvar aberrant maturations” (VAMs) [36]. These lesions arise out of lichenoid dermatitis and lack the basal atypia required for dVIN. Histological interpretation of dVINs and distinguishing them from VAMs and even HSILs remain challenging tasks with suboptimal interobserver agreement [37].

Clinically, dVINs usually appear as solitary, small, white (hyperkeratotic), or red lesions. Sometimes they are ill-defined and may be located in hairy areas (Figure 23a,b).

## 7. Conclusions

For detecting vulvar cancer early and referring the patients for appropriate treatment, an understanding of the importance of monitoring precursor lesions and recognizing the signs and symptoms of significant vulval disease is of utmost importance.

Diagnosis of vulvar intraepithelial neoplasia is delayed due to the absence of symptoms and specific features of the lesion. In general, these lesions can mimic lichen simplex chronicus, LS, psoriasis, Paget’s disease, or acute reactive vulvitis. Diagnosis can be achieved by biopsy only. A multicentric nature of a lesion requires multiple biopsies.

Distinguishing between HPV-related (vulvar HSIL) and HPV-independent (dVIN) precursors has important implications for treatment and prognosis. Human papillomavirus-independent SCC, which can rapidly develop from differentiated VIN, is less radiosensitive with higher disease-related mortality.

Clinical evaluation of all vulvar lesions, particularly high-grade ones, should be conducted very carefully, because an underlying early invasive squamous cancer, according to different reports, appears to be present in 20% of patients. Accurate mapping of biopsies and targeting them to the areas of most prominent changes is essential for accurate diagnosis and adequate treatment.

There is no screening for vulvar cancer. Thus, the only way to prevent vulvar cancer and the consequences of its treatment is to understand the natural history of vulvar precancer and recognizing its features on time.

## Figures and Tables

**Figure 1 cancers-14-01822-f001:**
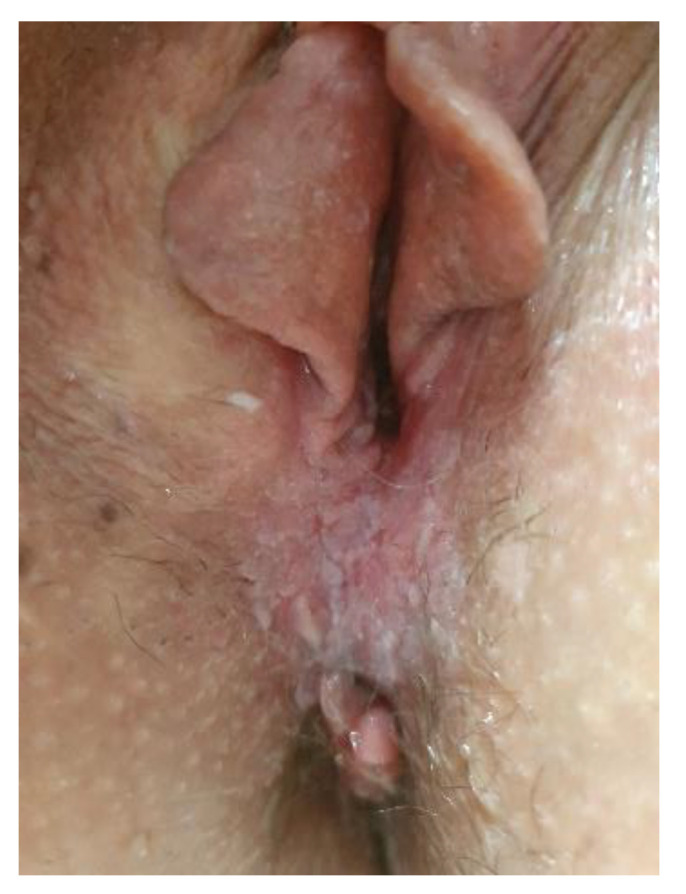
Extension of vulvar HSIL to the perianal area (after the application of acetic acid).

**Figure 2 cancers-14-01822-f002:**
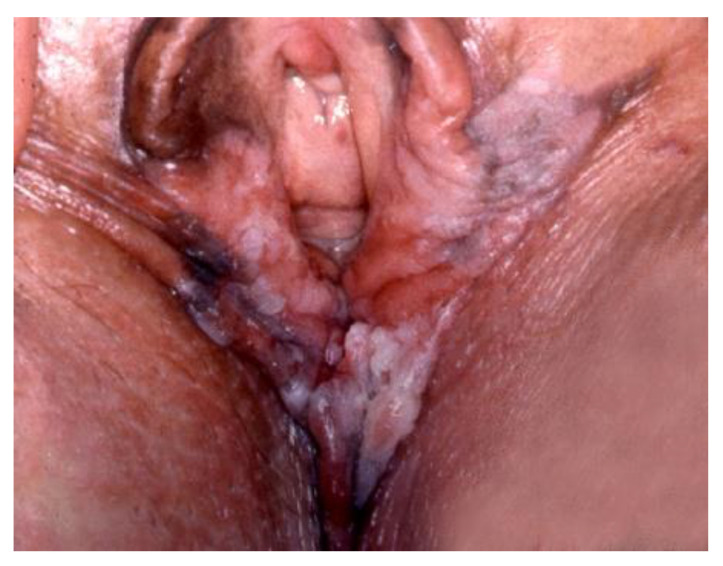
Biopsy-confirmed HSIL (after the application of acetic acid), with microinvasive cancer found after excision. Large lesions and/or multicentric lesions require multiple biopsies.

**Figure 3 cancers-14-01822-f003:**
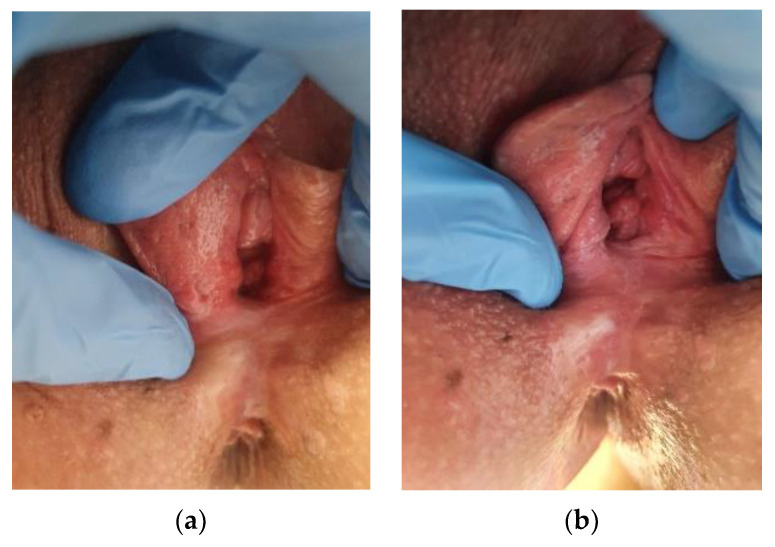
(**a**) Vulvar HSIL appearing as a red lesion; (**b**) after the application of Acetic acid.

**Figure 4 cancers-14-01822-f004:**
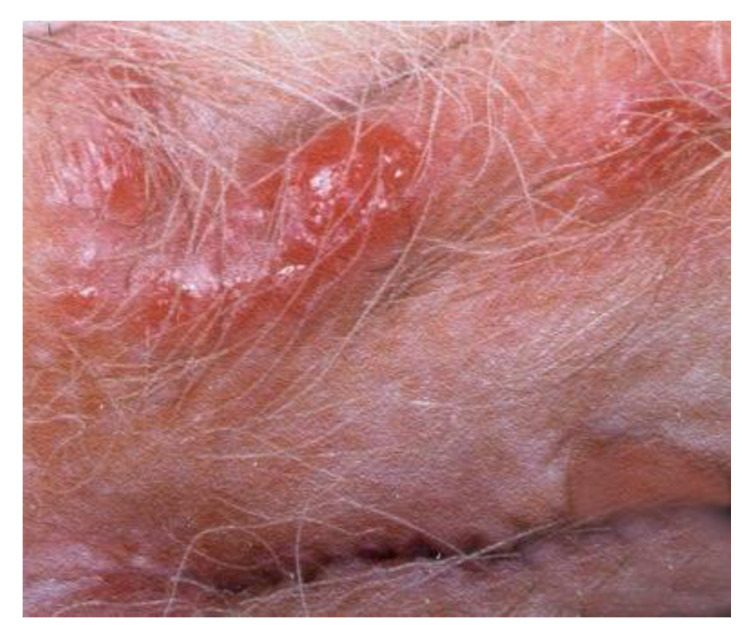
Folliculitis.

**Figure 5 cancers-14-01822-f005:**
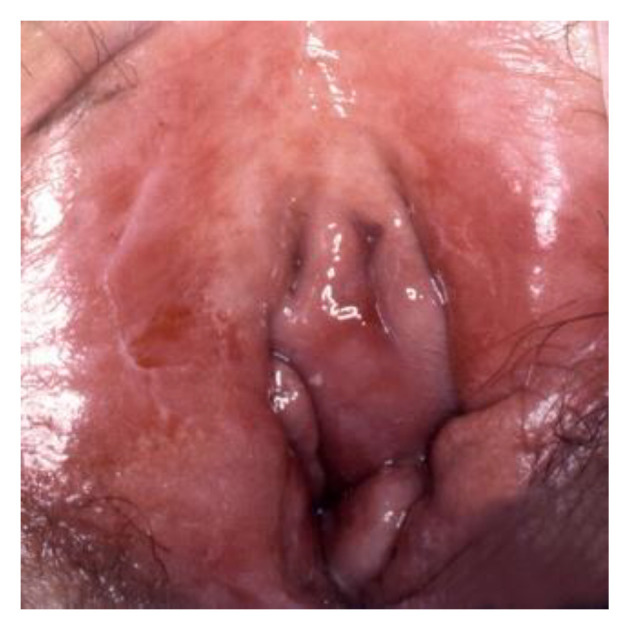
Lichen planus.

**Figure 6 cancers-14-01822-f006:**
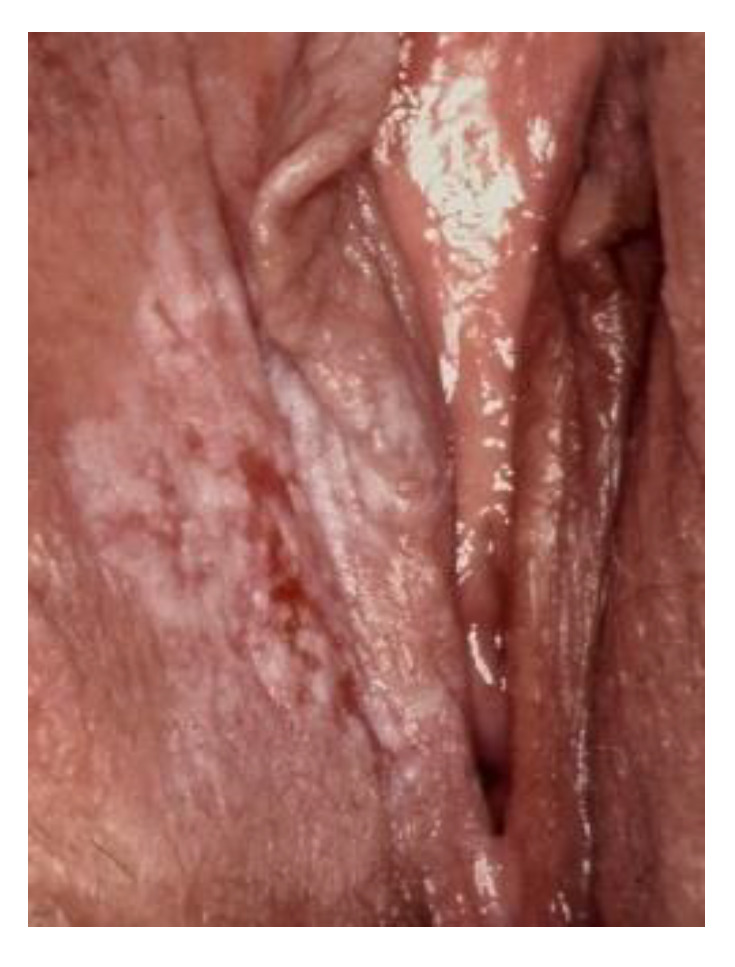
Paget’s disease.

**Figure 7 cancers-14-01822-f007:**
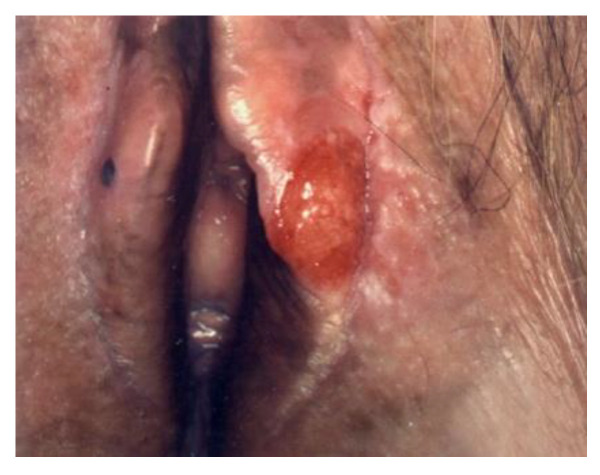
Invasive cancer.

**Figure 8 cancers-14-01822-f008:**
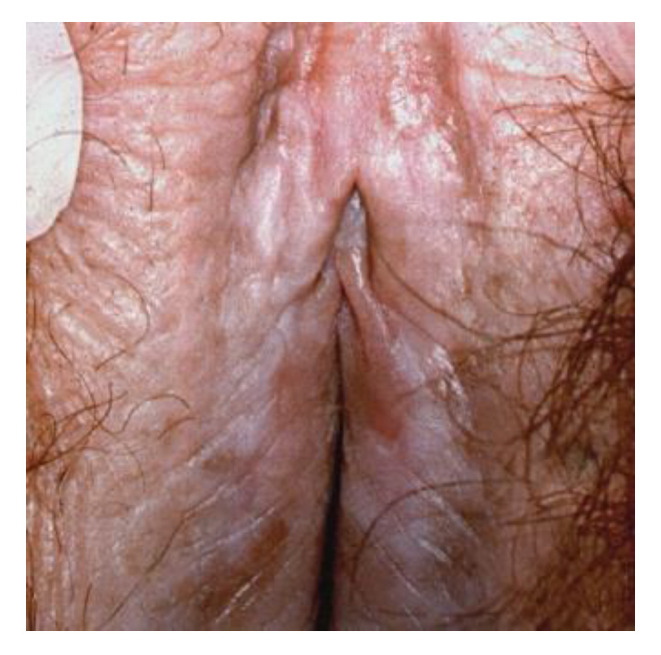
Lichen sclerosus.

**Figure 9 cancers-14-01822-f009:**
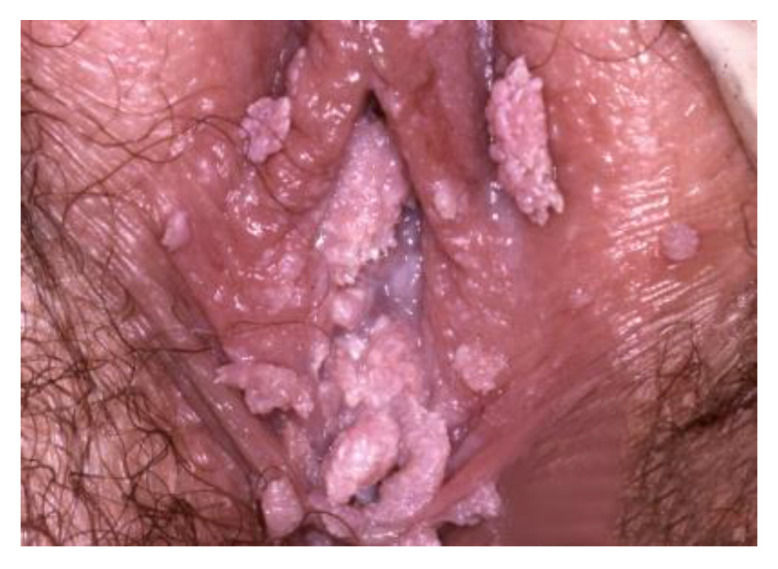
HPV infection.

**Figure 10 cancers-14-01822-f010:**
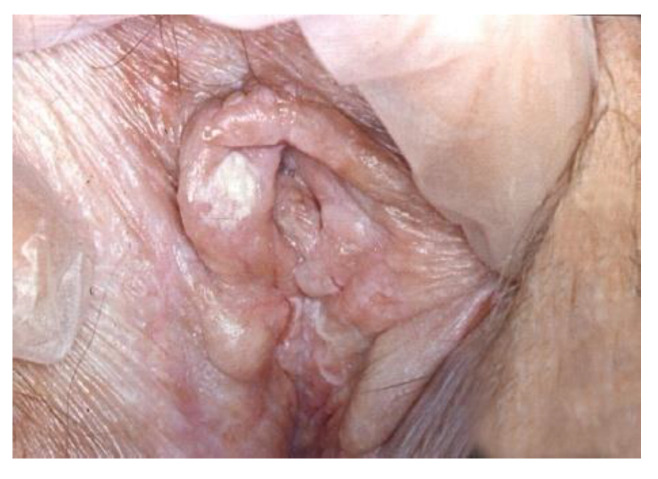
Vulvar HSIL. (before the application of Acetic acid).

**Figure 11 cancers-14-01822-f011:**
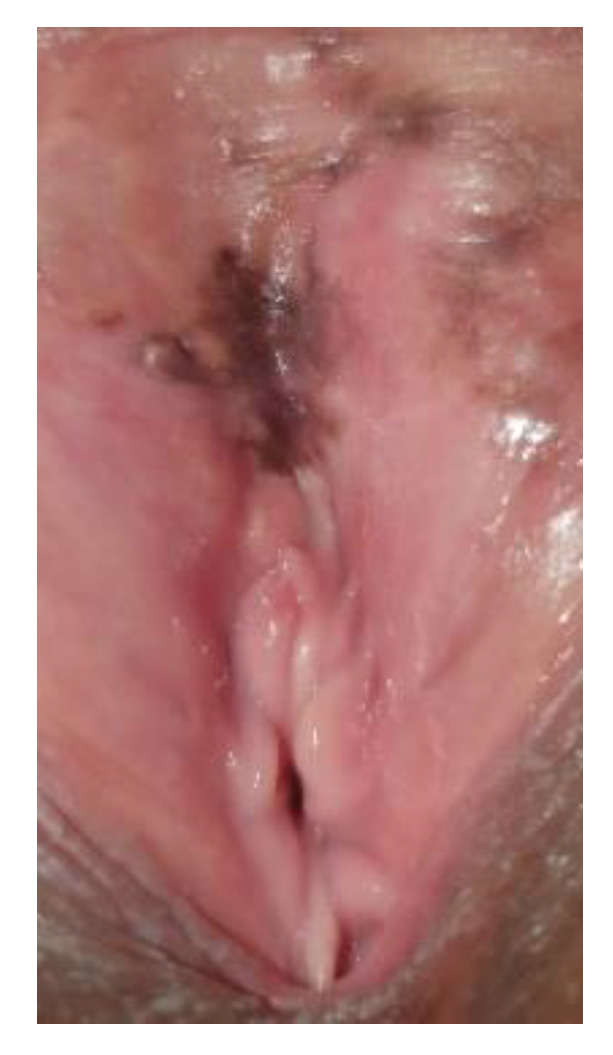
Hyperpigmentation.

**Figure 12 cancers-14-01822-f012:**
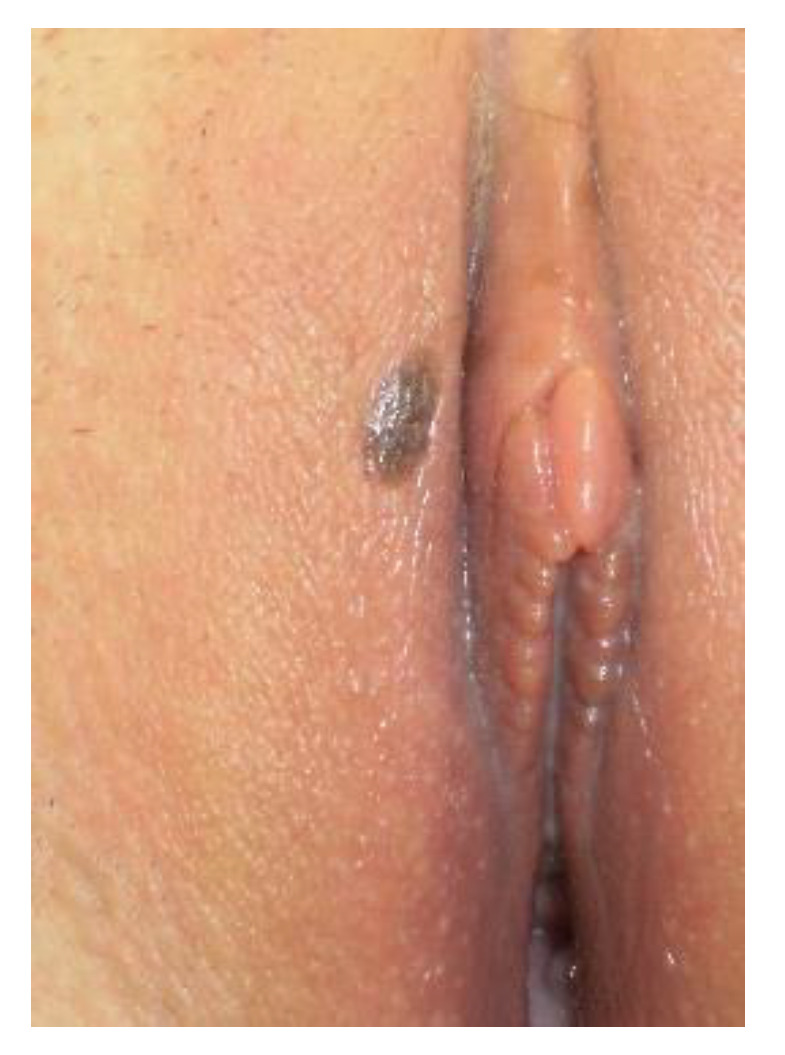
Nevus.

**Figure 13 cancers-14-01822-f013:**
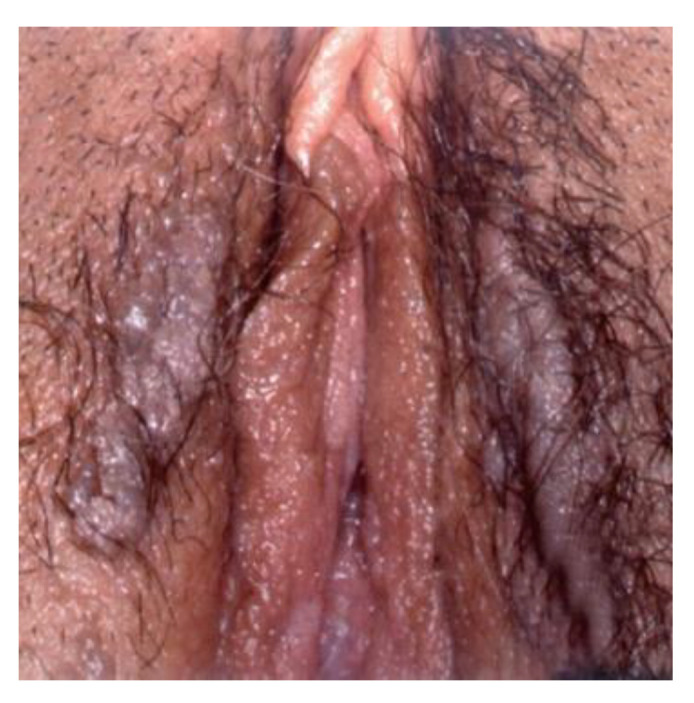
Vulvar HSIL.

**Figure 14 cancers-14-01822-f014:**
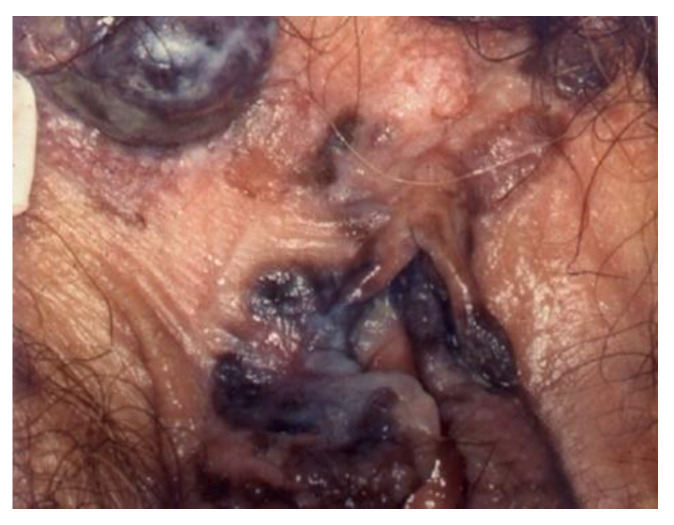
Melanoma.

**Figure 15 cancers-14-01822-f015:**
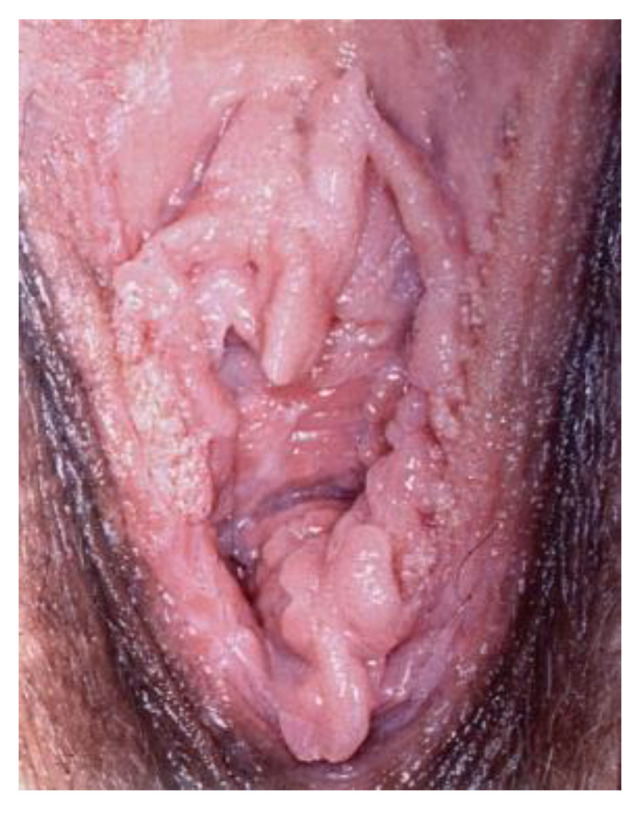
HPV infection.

**Figure 16 cancers-14-01822-f016:**
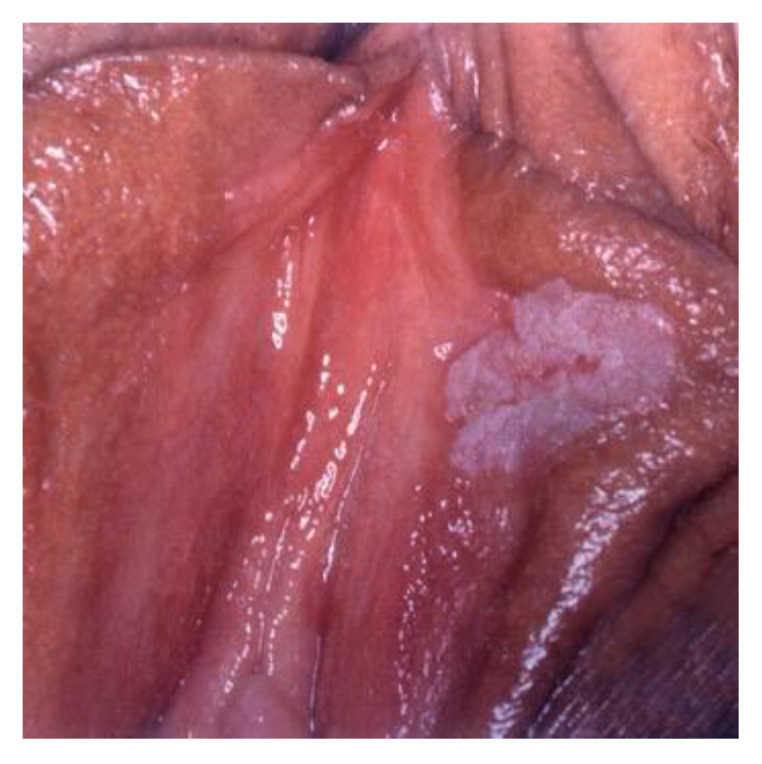
HSIL.

**Figure 17 cancers-14-01822-f017:**
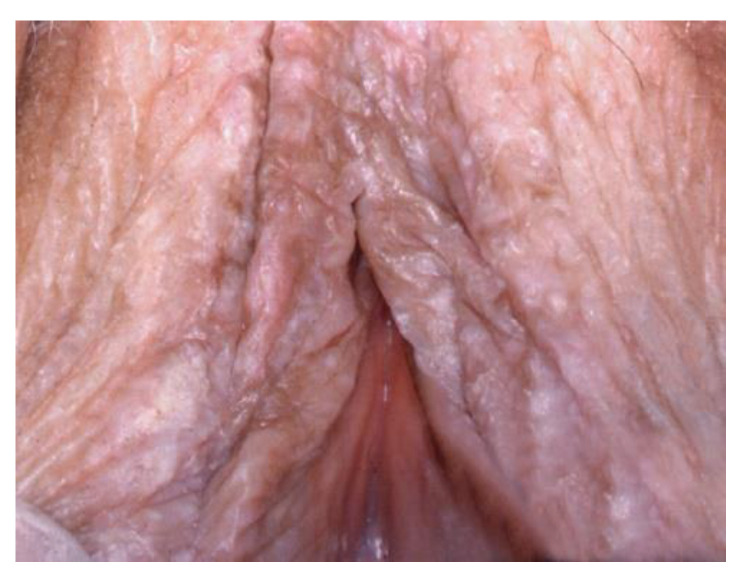
Lichenification.

**Figure 18 cancers-14-01822-f018:**
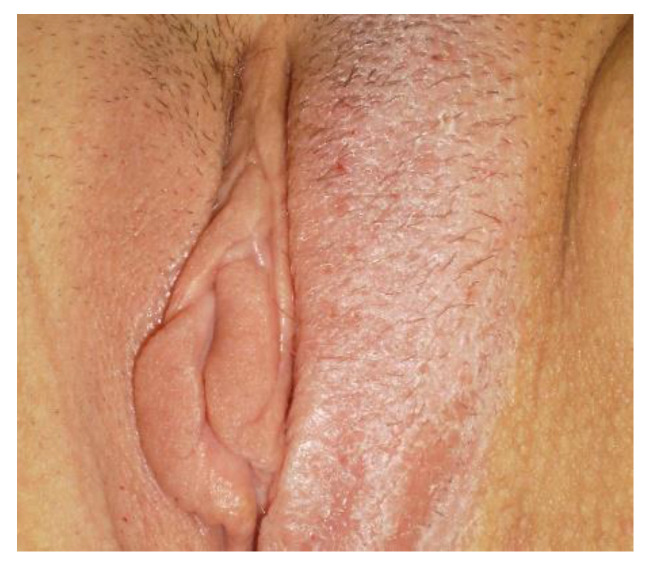
Psoriasis.

**Figure 19 cancers-14-01822-f019:**
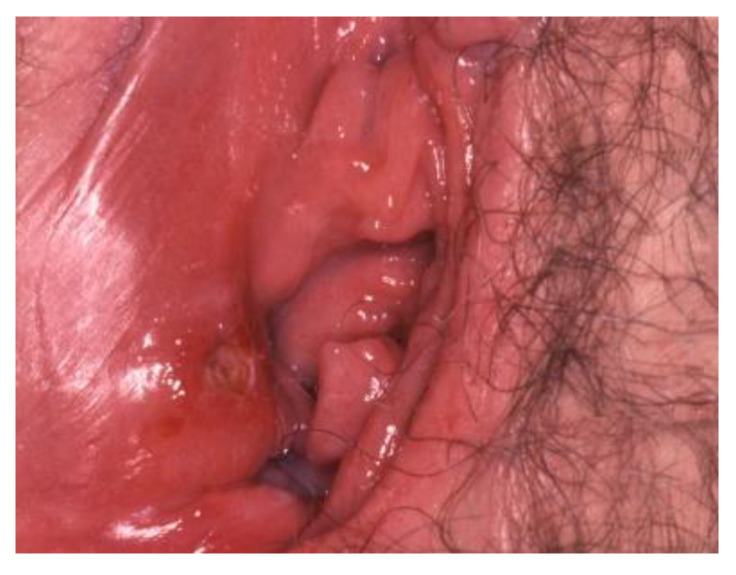
Ulcer in HIV-positive patient.

**Figure 20 cancers-14-01822-f020:**
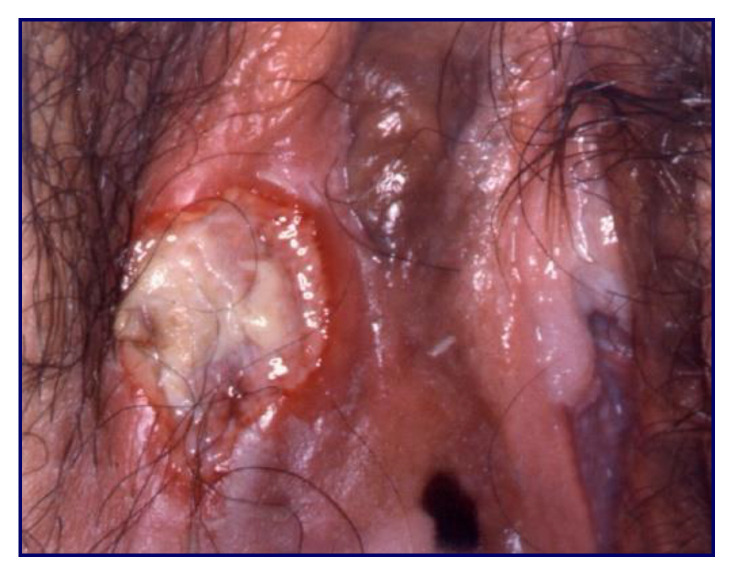
Behçet’s disease.

**Figure 21 cancers-14-01822-f021:**
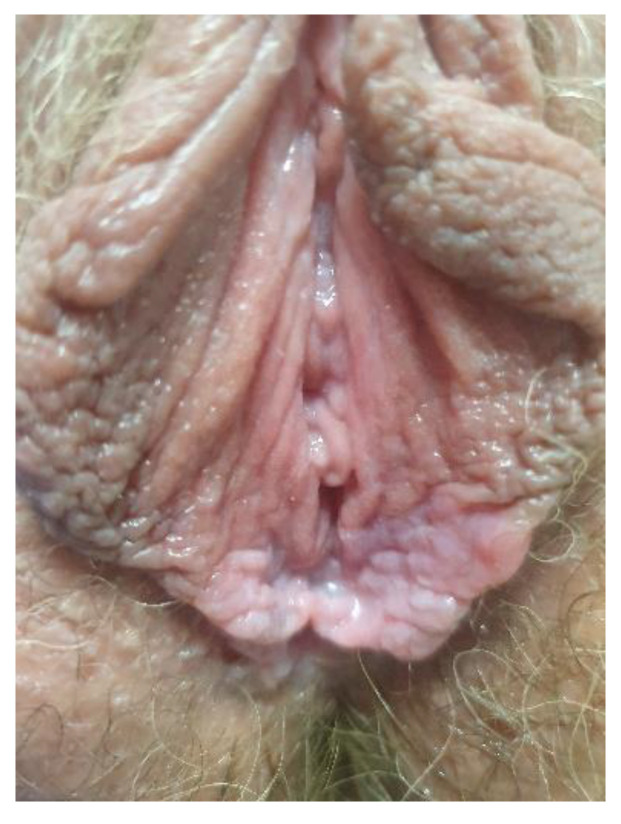
Vulvar HSIL. (after the application of acetic acid).

**Figure 22 cancers-14-01822-f022:**
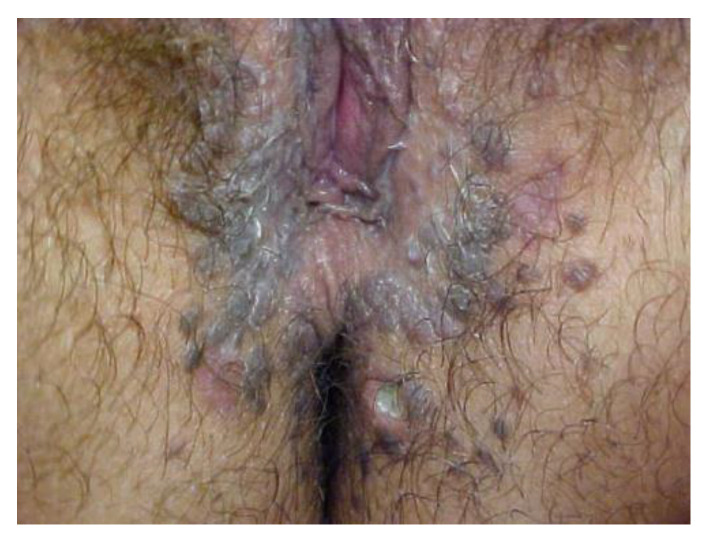
Pigmented HSIL.

**Figure 23 cancers-14-01822-f023:**
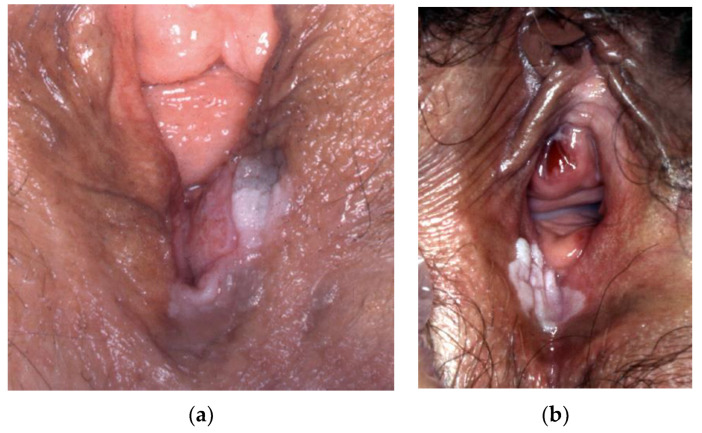
(**a**) dVIN, (**b**) dVIN.

**Table 1 cancers-14-01822-t001:** The 2011 IFCPC clinical/colposcopic terminology of the vulva (including the anus) *.

	Section		
Pattern			
Basic definitions	Various structures:		
	Urethra, Skene duct openings, clitoris, prepuce, frenulum, pubis, labia majora, labia minora, interlabial sulci, vestibule, vestibular duct openings, Bartholin duct openings, hymen, fourchette, perineum, anus, anal squamocolumnar junction (dentate line)		
	Composition:		
	Squamous epithelium: hairy/nonhairy, mucosa		
Normal findings	Micropapillomatosis, sebaceous glands (Fordyce spots), vestibular redness		
Abnormal findings	General principles: size in centimeters, location		
	Lesion type	Lesion color:	Secondary morphology:
	Macule	Skin-colored	Eczema
	Patch	Red	Lichenification
	Papule	White	Excoriation
	Plaque	Dark	Purpura
	Nodule	-	Scarring
	Cyst		Ulcer
	Vesicle		Erosion
	Bulla		Fissure
	Pustule		Wart
Miscellaneous findings	Trauma		
	Malformation		
Suspicion of malignancy	Gross neoplasm, ulceration, necrosis, bleeding, exophytic lesion Hyperkeratosis, with or without white, gray, red, or brown discoloration		
Abnormal colposcopic/other	Aceto-white epithelium, punctation, atypical vessels, surface irregularities		
Magnification findings	Abnormal anal squamocolumnar junction (notelocation about the dentate line)		

* Adapted from: Bornstein J, Sideri M, Tatti S, et al. (2011) [20] Terminology of the Vulva of the International Federation for Cervical Pathology and Colposcopy Journal of Lower Genital Tract Disease, Volume 16, Number 3, 2012, 290–295.

**Table 2 cancers-14-01822-t002:** Definitions of primary lesion types *.

Term	Definition
Macule	Small (<1.5 cm) area of color change; no elevation and no substance onpalpation
Patch	Large (>1.5 cm) area of color change; no elevation and no substance on palpation
Papule	Small (<1.5 cm) elevated and palpable lesion
Plaque	Large (>1.5 cm) elevated, palpable, and flat-topped lesion
Nodule	A large papule (>1.5 cm); often hemispherical or poorly marginated; may be located on the surface, within, or below the skin; nodules may be cystic or solid
Vesicle	Small (<0.5 cm) fluid-filled blister; the fluid is clear (blister: a compartmentalized, fluid-filled elevation of the skin or mucosa)
Bulla	A large (<0.5 cm) fluid-filled blister; the fluid is clear
Pustule	Pus-filled blister; the fluid is white or yellow
Eczema	A group of inflammatory diseases that are clinically characterized by the presenceof itchy, poorly marginated red plaques with minor evidence of microvesiculation and/or, more frequently, subsequent surface disruption
Lichenification	Thickening of the tissue and increased prominence of skin markings. Scale may or may not be detectable in vulvar lichenification. Lichenification may be bright red, dusky red, white, or skin-colored in appearance
Excoriation	Surface disruption (notably excoriations) occurring as a result of the “itch- cycle”
Erosion	A shallow defect in the skin surface; absence of some, or all, of the epidermis down to the basement membrane; the dermis is intact
Fissure	A thin, linear erosion of the skin surface
Ulcer	Deeper defect; absence of the epidermis and some, or all, of the dermis

* Adapted from Bornstein J, Sideri M, Tatti S, et al. 2011 [20] Terminology of the Vulva of the International Federation for Cervical Pathology and Colposcopy Journal of Lower Genital Tract Disease, Volume 16, Number 3, 2012, 290–295.

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
