# Peer review of "Early Diagnostics of Vulvar Intraepithelial Neoplasia"

_cancers, 2022, doi:10.3390/cancers14071822_

Round 1

Reviewer 1 Report

The authors give a good overview of vulvar lesions in general and vulvar HSIL in particular. However the authors are inconsistent with the terms “vulvar HSIL”, “VIN”, “HSIL of the vulvar”, “high-grade VIN”. The authors did not include the WHO 2014 classification in their review. 

  • Simple summary p1 l12 The authors state that there are hormonal and systemic disturbances as a spectrum of vulvar disorder. Please give examples.
  • Abstract: The authors state that vulvar lesions differ in their clinical appearance. However tha common symptom for vulvar lesion is pruritus in many cases regardless of the underlying histology. The authors should consider this.
  • P4 line137: The use of acetic acid can be essential in detecting HISL of the vulvar as some lesions are not detected by inspection alone e.g. VIN2. Nevertheless almost every use of acetic acid will lead to a reaction. The authors need to be more specific at this section.
  • Biopsy: each lesions resistant to conservative therapy such as e.g.steroids, antibiotics, estrogen, must have a biopsy performed, please consider
  • P5 line 174: Please state: Bornstein et al. subdivided lesion color in four different colors, why have the authors subdivided them into three colors. What is the reason for having the two colors in one group?
  • P5 line the authors give “VIN” as an example for red lesions. This is somehow misleading as most VIN are visuable after applying acetic acid and in that case are of white appearance
  • Figure 7 Please consider a different picture as an example for HPV infection. This picture is not a good example especially for inexperienced clinicians.
  • Figure 8: is this high-grade VIN HPV-dependent or independent? Is that picture taken after applying acetic acid to the vulva? The authors should consider a different picture for a high-grade VIN and state clearly if it is HPV-dependent or independent.
  • Figure 9: The legend seems to be incomplete: “Hyper”
  • Please consider a picture of a malignant melanoma
  • The legends of the different figures are inconsistent figure 8: high grade VIN; figure 11: Vulvar HSIL; figure 14: HSIL; be more consistent, as this can easily confuse the inexperienced reader. There are also different terms used for vulvar HSIL throughout the text, please be more accurate.
  • P8 line 236: please state a subliclinical HPV infections is histological identical with vulvar LSIL see WHO 2014 classification
  • Figure 13 is not a good example for HPV infection
  • In general, the authors should quote if the pictures are taken before or after applying acetic acid as this is an important information for the reader
  • The authors neglect the WHO 2014 classification in their overview, this should be considered as pathologists classify VIN according to the WHO 2014 classification.
  • P9 line 291-293: what is definition of young and old women?
  • P9 line 297, another important risk factor is a history of HSIL of cervix, vagina or vulva
  • P9 line 327 what is meant by the abbreviation “VAM”?
  • P9 line 345 “Exact mapping of biopsies”, this sentences seams to incomplete

Author Response

Respected Reviewer,

We would like to thank you for the time and effort they made to read so thoroughly our article and give useful and detailed comments and advice. Before giving our explanations, I must say that I am afraid I sent to editor the version “just to be finished”. This is why you noticed mistakes at some points.

Just one note more… after corrections and adding new pictures and references, the numbers of figures and citations are changed.

Reviewer 1

  • The authors give a good overview of vulvar lesions in general and vulvar HSIL in particular. However the authors are inconsistent with the terms “vulvar HSIL”, “VIN”, “HSIL of the vulvar”, “high-grade VIN”. The authors did not include the WHO 2014 classification in their review. 

Thank you for this observation. WHO classification 2014 is the same as the LAST 2012 and ISVVD 2015. We agreed to use the SIL terminology: LSIL, HSIL and dVIN . Appropriate corrections are made

  • Simple summary p1 l12 The authors state that there are hormonal and systemic disturbances as a spectrum of vulvar disorder. Please give examples.

Thank you for comment. Since the number of words in abstract is limited, we did not explain more the vulvar manifestations of hormonal and systemic disturbances. In the text we provided illustrations for some systemic conditions such as Bechet’s disease and Psoriasis.

  • Abstract: The authors state that vulvar lesions differ in their clinical appearance. However the common symptom for vulvar lesion is pruritus in many cases regardless of the underlying histology. The authors should consider this.

We used the formulation “lesions differ in their clinical appearance” meaning that not the symptoms, but their clinical appearance differ. We, however, consider this and instead of “appearance” the term “features” will be used. Also, the importance of pruritus as a common symptom is mentioned (P3, lines 93-95). The number of words in the abstract is also limited.

  • P4 line137: The use of acetic acid can be essential in detecting HISL of the vulvar as some lesions are not detected by inspection alone e.g. VIN2. Nevertheless almost every use of acetic acid will lead to a reaction. The authors need to be more specific at this section.

We agree, the use of acetic acid leads to reaction. However, most of high-grade VIN are visible by visual inspection. As identified, the reaction to Acetic acid is not specific and may appear in conditions which cannot be categorized as HSIL, however, it can be useful to identify the extent of a lesion or to identify a subclinical lesion for biopsy. Corrections are made in regard to the effects of Acetic acid. (P5, lines 161-166).

  • Biopsy: each lesions resistant to conservative therapy such as e.g.steroids, antibiotics, estrogen, must have a biopsy performed, please consider

We think that all lesions suspicious to high grade VIN should be biopsied. Conservative treatment without histologic confirmation is not preferable. Nevertheless, the correction is made considering you comment. Text “Lesions resistant to local medical treatment” was added as well as second reviewr’s suggestion, to the indications for biopsy  (P5, lines 183)

  • P5 line 174: Please state: Bornstein et al. subdivided lesion color in four different colors, why have the authors subdivided them into three colors. What is the reason for having the two colors in one group?

You are right. There is no reason for grouping skin colored and red into the one category except the fact that high grade vulvar lesions are usually not skin colored.  Correction is made

  • P5 line the authors give “VIN” as an example for red lesions. This is somehow misleading as most VIN are visible after applying acetic acid and in that case are of white appearance

Yes, most VIN are visible after applying acetic acid, but since application of Acetic acid and looking through the colposcope is not initial procedure for examining vulva, red lesion is the one which should induce application of acetic acid. HSIL may present just as a red surface. We provide the illustration (Figure 3a and b) and emphasize the need of applying Acetic acid.

  • Figure 7 Please consider a different picture as an example for HPV infection. This picture is not a good example especially for inexperienced clinicians.

We chose this picture thinking that that usual appearance of HPV infection should be known to all clinicians. The picture has been replaced with the new one. Corrected.

  • Figure 8: is this high-grade VIN HPV-dependent or independent? Is that picture taken after applying acetic acid to the vulva? The authors should consider a different picture for a high-grade VIN and state clearly if it is HPV-dependent or independent.

High grade SIL of vulva (usuall type of VIN) is HPV dependent. This picture was taken before the application of Acetic acid. Histology was VIN2, HPV dependent. Satellite lesions visible in posterior fourchette area. I agree that on the first look this lesion resembles more to dVIN. We replaced the illustration with the new one. 

  • Figure 9: The legend seems to be incomplete: “Hyper” Corrected (hyperpigmentation)

  • Please consider a picture of a malignant melanoma

Picture of melanoma was already included (Figure 14)

  • The legends of the different figures are inconsistent figure 8: high grade VIN; figure 11: Vulvar HSIL; figure 14: HSIL; be more consistent, as this can easily confuse the inexperienced reader. There are also different terms used for vulvar HSIL throughout the text, please be more accurate.

Thank you again for so detailed review and valuable comments. We accept this remark with gratitude

  • P8 line 236: please state a subliclinical HPV infections is histological identical with vulvar LSIL see WHO 2014 classification Corrected

  • Figure 13 is not a good example for HPV infection

Figure 13 is intended to present subclinical HPV infection. Agree, not the best example. Replaced with the new one for subclinical HPV

  • In general, the authors should quote if the pictures are taken before or after applying acetic acid as this is an important information for the reader

We did it where applicable. The information about pre- or post- Acid acetic was not provided because most of the high grade lesions were clearly visible by naked eye

  • The authors neglect the WHO 2014 classification in their overview, this should be considered as pathologists classify VIN according to the WHO 2014 classification.

Comment accepted

  • P9 line 291-293: what is definition of young and old women?

Great question ?. We added the data on incidence and prevalence of HPV infection, mentioning the age of 35 considered young in the context of HPV related vulvar lesions (P12, line 345-350)

  • P9 line 297, another important risk factor is a history of HSIL of cervix, vagina or vulva. Added

  • P9 line 327 what is meant by the abbreviation “VAM”?

Vulvar aberrant maturation - VAM. Mentioned in the sentence before. In the final version the full name was provided together with abbreviation

  • P9 line 345 “Exact mapping of biopsies”, this sentences seams to incomplete

This is another mistake which was corrected in the final version. The problem is that both texts were saved under the same date and I did not take care about the exact time. So a few hours older version was sent as a final. Correction made

Reviewer 2 Report

The article is readable and easy to understand;

The aim of the article is clear and well written, some minor comments:

line 26 Colposcopy of the vulva (vulvoscopy)

"The term vulvoscopy should not be intended as the colposcopic examination of the vulva. The term vulvoscopy can be retained but should be intended as a composite diagnostic act composed of careful naked-eye and low-power magnified examination carried out by gynecologists or colposcopists with interdisciplinary (gynaecology, dermatology and pathology) knowledge and experience." (Micheletti 2011)

Please consistently try to underline this point of view "not a colposcopy of the vulva"

 line 68 Studies confirm elevated prevalence of anal intraepithelial neoplasia in women with HPV related precancer and cancer [5]. The evaluation of the anal canal should be performed only by clinicians with adequate training [6].

The two sentences are not very clear ..the message is: do not perform anoscopy during a vulvoscopy if you are expert in this field?

The message is not so clear; in other word, if I am expert also in anal precancers should I perform an anoscopy too? All women with VIN should be submitted to anoscopy? please clarify. You should consider an approach like "there is no indication to perform other exams" [ such as anoscopy, partner's penoscopy, flexible nasal endoscopy (ENT)..] ? 

line 80-2 I don't understand the reason of the italics, is it necessary?

line 108 This examination can be unpleasant and even painful for some women. "Why watching should be painful?" may be "embarrassing"? this could be an equivocal message for neophytes of vulvoscopy.

line 136 It should be, however, remebered that such liberal use of acetic acid may be misleading in provoking non‐specific aceto‐white reactions [4].

Acetowhitening of the vulva has high sensitivity but low specificity as a predictor of high grade vulvar intraepithelial neoplasia. The absence of acetowhite lesion can reassure that high grade vulvar lesion is absent; please  underline that in case of routinary use specifity of the test is about 40%. (santoso 2015)

line 152 consider adding "therapy failure"

line 299 please report also information about frequency of invasive cancer in surgically excised vulvar lesions with intraepithelial neoplasia.

Rare diseases of the vulva, such as Zoon vulvitis, histocytosis, etc are not mentioned a short paragraph should be interesting.

Author Response

Respected Reviewer,

We would like to thank you for the time and effort they made to read so thoroughly our article and give useful and detailed comments and advice. Before giving our explanations, I must say that I am afraid I sent to editor the version “just to be finished”. This is why you noticed mistakes at some points.

Just one note more… after corrections and adding new pictures and references, the numbers of figures and citations are changed.

Reviewer 2

line 26 Colposcopy of the vulva (vulvoscopy)

"The term vulvoscopy should not be intended as the colposcopic examination of the vulva. The term vulvoscopy can be retained but should be intended as a composite diagnostic act composed of careful naked-eye and low-power magnified examination carried out by gynecologists or colposcopists with interdisciplinary (gynaecology, dermatology and pathology) knowledge and experience." (Micheletti 2011)

Please consistently try to underline this point of view "not a colposcopy of the vulva"

Thank you very much for this comment. We have already cited Micheletti, but we shall repeat and emphasize the difference between colposcopy and vulvoscopy.

 line 68 Studies confirm elevated prevalence of anal intraepithelial neoplasia in women with HPV related precancer and cancer [5]. The evaluation of the anal canal should be performed only by clinicians with adequate training [6].

The two sentences are not very clear ..the message is: do not perform anoscopy during a vulvoscopy if you are expert in this field?

The message is not so clear; in other word, if I am expert also in anal precancers should I perform an anoscopy too? All women with VIN should be submitted to anoscopy? please clarify. You should consider an approach like "there is no indication to perform other exams" [ such as anoscopy, partner's penoscopy, flexible nasal endoscopy (ENT)..] ? 

We changed this sentence into “If evaluation of the anal canal is to be performed the patients should be referred to clinicians with adequate training”.(P3, lines 112-113)

line 80-2 I don't understand the reason of the italics, is it necessary?

Sorry, this is one of my mistakes. In the previous version the names of different anatomical parts were written in Latin language

line 108 This examination can be unpleasant and even painful for some women. "Why watching should be painful?" may be "embarrassing"? this could be an equivocal message for neophytes of vulvoscopy.

This is another mistake which was corrected in the final version. As I explained before, the problem is that I had two texts saved under the same date and I did not take care about the exact time. So a few hours older version was sent as a final. Corrections were mad ein the section Technique of vulvoscopy.

line 136 It should be, however, remebered that such liberal use of acetic acid may be misleading in provoking non‐specific aceto‐white reactions [4].

Acetowhitening of the vulva has high sensitivity but low specificity as a predictor of high grade vulvar intraepithelial neoplasia. The absence of acetowhite lesion can reassure that high grade vulvar lesion is absent; please  underline that in case of routinary use specifity of the test is about 40%. (santoso 2015)

Thank you for help. Sensitivity and specificity of Acetic acid added (P5, line 163-168)

line 152 consider adding "therapy failure"  Therapy failure will be added

line 299 please report also information about frequency of invasive cancer in surgically excised vulvar lesions with intraepithelial neoplasia.

Important comment. We did not mention it because other authors were writing article about treatment. But indeed, this should be emphasized. Corrections made (P5, line 187-189, P12 line 337-40)

Rare diseases of the vulva, such as Zoon vulvitis, histocytosis, etc are not mentioned a short paragraph should be interesting.

Rare coditions are without doubt very challenging and important to recognize. However, the limit of words in the text as well as the focus on squamous precancerous lesions. Nevertheless, we mentioned some of these conditions, when commenting the features of vulvar intraepithelial lesions.

Round 2

Reviewer 1 Report

The authors have corrected the mentioned suggestions as requested.